# *MoMCP1*, a Cytochrome P450 Gene, Is Required for Alleviating Manganese Toxin Revealed by Transcriptomics Analysis in *Magnaporthe oryzae*

**DOI:** 10.3390/ijms20071590

**Published:** 2019-03-29

**Authors:** Yi Wang, Qi Wu, Lina Liu, Xiaoling Li, Aijia Lin, Chengyun Li

**Affiliations:** 1State Key Laboratory for Conservation and Utilization of Bio-Resources in Yunnan, Yunnan Agricultural University, Kunming 650201, China; wyi_0114@163.com (Y.W.); wuqiynau@163.com (Q.W.); handyliu@126.com (L.L.); lxl_1806@163.com (X.L.); laj7411@126.com (A.L.); 2College of Science, Yunnan Agricultural University, Kunming 650201, China; 3Agricultural Environment and Resources Institute, Yunnan Academy of Agricultural Sciences, Kunming 650205, China; 4Kunming Edible Fungi Institute of All China Federation of Supply and Marketing Cooperatives, Kunming 650223, China

**Keywords:** *Magnaporthe oryzae*, manganese toxin, transcriptomics

## Abstract

Manganese, as an essential trace element, participates in many physiological reactions by regulating Mn associated enzymes. *Magnaporthe oryzae* is a serious pathogen and causes destructive losses for rice production. We identified a cytochrome P450 gene, *MoMCP1*, involving the alleviation of manganese toxin and pathogenicity. To identify the underlying mechanisms, transcriptomics were performed. The results indicated that many pathogenicity related genes were regulated, especially hydrophobin related genes in ∆*Momcp1*. Furthermore, the Mn^2+^ toxicity decreased the expressions of genes involved in the oxidative phosphorylation and energy production, and increased the reactive oxygen species (ROS) levels, which might impair the functions of mitochondrion and vacuole, compromising the pathogenicity and development in ∆*Momcp1*. Additionally, our results provided further information about Mn associated the gene network for Mn metabolism in cells.

## 1. Introduction

Manganese, as an essential trace element, plays a vital role in all kingdoms of life, and there are a diverse range of enzymes utilizing Mn^2+^ as a key cofactor to activate enzyme function, which help the cell resist oxygen damage [1], recover the activity of proteins and initiate the gene expression [2]. Mn can be distributed in many organelles, such as the nucleus, mitochondria, cytosol, golgi, and vacuole. However, the excess accumulation of manganese can be toxic to cells. In humans, the exposure to Mn^2+^ can result in manganism, which is associated with Parkinson’s disease [3]. In plants, the typical Mn^2+^ toxicity symptoms are characterized as the brown spots on the mature leaves due to increasing peroxidase activity mediated by phenolics [4]. Furthermore, there are many Mn containing fungicides, such as maneb and mancozeb. Thus, understanding the relationship between the biological process and Mn^2+^ homeostasis helps uncover the mechanism of Mn^2+^ in life. 

Manganese related transporters have been well identified and help to mediate the Mn^2+^ homeostasis. In rice, transporters involved in the Mn^2+^ uptake are OsNramp5 and OsMTP9. Nramp5 is polarly localized at the distal side of both exodermis and endodermis cells [5], while OsMTP9 is localized at the proximal side of these cell layers [6]. At high Mn^2+^ concentrations, two transporters, OsYSL6 and OsMTP8.1, are involving in the detoxification in rice leaves. OsYSL6 seems to transport Mn^2+^–nicotianamine from the apoplast to symplast [7], and OsMTP8.1 transports Mn^2+^ into the vacuoles for sequestration [8]. OsMTP11 was identified as a trans-Golgi network localized transporter, and showed a significant relocalization of OsMTP11 to the plasma membrane at high levels of extracellular Mn^2+^ in tobacco epidermal cells [9]. In yeast, Smf1p, a cell surface transporter, transports the Mn^2+^ into cells involving oxidative stress [10]. Smf2p is located in Golgi-like vesicles and shows a significant impact on Mn^2+^ homeostasis through regulating the Mn-SOD activity [11]. Moreover, yeast also possesses the transporters Pmr1p and Ccc1p to transport excess Mn^2+^ in cells. Pmr1p, a P-type Ca^2+^- and Mn^2+^-transporting ATPase that is localized in the Golgi membrane, transport excess Mn^2+^ into the Golgi and excrete via the secretory pathway [12]. Ccc1p, a vacuolar manganese transporter, helps to sequester excess manganese in the vacuole [13]. 

During the interaction between pathogens and hosts, competing for manganese is a strategy for both sides [14]. To defend the pathogen infection, a host generating ROS could oxidase Fe^2+^ and inactivate many Fe dependent enzymes in pathogens through a Fenton reaction, which increases the cellular freely diffusible hydroxyl radical (OH•) and enhances the cell damage. Therefore, pathogens need Mn^2+^ to rescue enzyme activities and detoxify ROS. Corbin et al. (2008) found neutrophil-derived calprotectin inhibited *Staphylococcus aureus* growth through the chelation of nutrient Mn^2+^ and Zn^2+^ in tissue abscesses, while the staphylococcal proliferation was enhanced in these metal-rich abscesses [15]. Furthermore, Nramp1 is responsible for removing Mn^2+^ from the phagosome to restrict microbial access to Mn^2+^ in hosts [16]. To cope with manganese starvation, many Mn^2+^ import transporters that are involved in pathogenesis were identified in bacterial pathogens, such as Mn^2+^ H-family or ABC-family [17]. Moreover, Radin et al. (2016) established a global staphylococcal virulence regulator, ArlRS, that can help *S. aureus* resist calprotectin-induced manganese starvation through altering the cellular metabolism [18]. The type VI secretion system in *Burkholderia thailandensis* has been proved to not only be related to bacterial virulence, but also to be involved in the transport of Mn^2+^ to scavenge oxidative stress [19]. Therefore, the competition for Mn^2+^ could not be a neglected tactic between the pathogens and hosts. Importantly, recent studies reported that manganese increased the sensitivity of the cGAS-STING pathway for double-stranded DNA and helps host against DNA viruses, which provided the direct relationship between Mn^2+^ and innate immunity [20]. 

*Magnaporthe oryzae* caused a devastating rice blast disease and resulted in yield losses in the global rice production. During the interaction between rice and blast fungus, the induction of H_2_O_2_ is regarded as a resistance strategy [21,22]. For rice blast fungus, the modulation of cellular Mn might be involved in ROS elimination and conquer the host defense in rice, but the Mn associated genes were hardly reported. We identified a cytochrome P450 gene regulated by Mn^2+^ and found that this gene is required for pathogenicity and excessive Mn^2+^ tolerance. Transcriptomics results showed that Mn^2+^ toxicity decreased the expressions of genes involved in the oxidative phosphorylation and energy production, increased the ROS levels which might impair the functions of mitochondrion and vacuole, compromising the pathogenicity and development in ∆*Momcp1*. 

## 2. Results

### 2.1. ∆Momcp1 Mutant Is More Sensitive to Mn^2+^

MoMCP1 showed 93% identities of amino acid sequences to a fragment of cytochrome P450 oxidoreductase, compared with the reference genome 70-15 (Appendix A). We also searched this gene in 177 published genomes of *M. oryzae*, and this gene was found in 22 genomes with 100% identities according to DNA sequences (Appendix A). The results indicate that *MoMCP1* is not distributed in all blast strains. A ∆*Momcp1* deletion mutant was obtained by the replacement of a *MoMCP1* coding sequence with a hygromycin resistance cassette and identified by Polymerase Chain Reaction (PCR) and Sanger sequencing (Appendix A). However, there were no significant differences between wild type (WT) and ∆*Momcp1* in the colony appearance or conidial production (Figure 1A,E). 

In order to determine whether the ∆*Momcp1* mutant was sensitive to Mn^2+^, we compared the colony growth under the Mn^2+^ treatment. There were no significant differences in the inhibitory rates between the WT and ∆*Momcp1* mutant, but the thinner mycelium was observed in the mutant (Figure 1A,C). Therefore, we used the liquid complete medium (CM) culture, adding Mn^2+^, to assess the mycelium development and dry weight. The ∆*Momcp1* mutant showed less and smaller mycelium under the Mn^2+^ treatment compared with WT. Furthermore, the dry weight of the ∆*Momcp1* mutant was significantly reduced (Figure 1B,D). Similarly, we found that the conidial production ability in the ∆*Momcp1* mutant was decreased significantly under Mn^2+^ stress (Figure 1E). These results indicated that the ∆*Momcp1* mutant is more sensitive to Mn^2+^.

### 2.2. MoMCP1 Is Involved in the Pathogenicity of Rice

The pathogenicity of the ∆*Momcp1* mutant was performed by conidia spay method. The lesions caused by the mutant showed fewer numbers and smaller areas compared with WT (Figure 2A,B). We used real-time quantitative PCR (RT-qPCR) to assess the expressions of defense related genes in rice with the ∆*Momcp1* mutant inoculation (Figure 2C). The expressions of *OsEDS1* and *OsSID2* were significantly increased in rice infected with the ∆*Momcp1* mutant at 24 hpi (hour post inoculation). Similarly, the expression of the other two genes, *PR1a* and *PR10a*, was also increased significantly at 48 hpi and 72 hpi. Thus, the expressions of defense related genes were increased in rice after the ∆*Momcp1* mutant infection. To determine the direct effect of Mn^2+^ on fungal infection, we performed the punch inoculation with two strains under Mn^2+^ treatment on detached leaves. The length and biomass of the two strains were both decreased (Figure 2D–F), while the pathogenicity of ∆*Momcp1* was significantly reduced compared with WT. These results showed that excessive Mn^2+^ disrupts the fungal pathogenicity, especially for ∆*Momcp1*.

### 2.3. MoMCP1-GFP Fusion Protein Is Only Observed in Hypha

To explore the expression pattern of MoMCP1, we constructed a MoCMP1-GFP fusion protein with a strong constitutive promoter *RP27* to observe the location of MoMCP1 during the fungal development. We found that the recombined green fluorescent protein (GFP) was only detected in the hypha, while there is no signal in the conidia (Figure 3A,B). Furthermore, we also found that there was no appressorium formation, and strong green fluorescent signals were observed in the branched hypha germinated from the conidia (Figure 3C). These results indicated that MoMCP1 expressed in the hypha specifically, and the overexpression of *MoMCP1* impaired the appressorium formation.

### 2.4. Excessive Mn^2+^ Increased Intracellular Content of H_2_O_2_ and Decreased ROS Related Enzymes in ∆Momcp1 Mutant

According to previous reports, the excessive accumulation of Mn^2+^ could induce ROS toxicity. Therefore, we determined the H_2_O_2_ contents and the activities of many antioxidant enzymes such as superoxide dismutase (SOD), peroxidase (POD), and catalase (CAT) under Mn^2+^ stress in the WT and ∆*Momcp1* mutant. The results showed that the content of H_2_O_2_ was reduced in the WT, but increased in the ∆*Momcp1* mutant (Figure 4A). The SOD activities were both increased significantly when treated with Mn^2+^ in both strains, but the level of SOD in the WT was much higher than in the ∆*Momcp1* mutant (Figure 4B). However, decreased CAT activities were found in both the WT and ∆*Momcp1* mutant (Figure 4C), and there were no significant changes of POD activities under the Mn^2+^ treatment (Figure 4D). Therefore, excessive Mn^2+^ generating a lower activity of SOD and more content of H_2_O_2_ might be the reason for the Mn^2+^ sensitivity in the ∆*Momcp1* mutant.

### 2.5. Excessive Mn Affected the Contents of Cu, Zn, Fe, and Ca in M. oryzae

We used Inductively Coupled Plasma Optical Emission Spectrometry (ICP-OES) to determine the changes of the intracellular metal contents including Mn, Cu, Zn, Fe, and Ca under the Mn^2+^ treatment. However, the contents of Cu, Zn, Ca and Mn were significantly increased both in the WT and ∆*Momcp1* mutant (Figure 5A,C–E), and the Mn content in the WT was higher than in the ∆*Momcp1* mutant, implying that the sensitive strain possessed a smaller amount of Mn. Meanwhile, the content of Fe in the WT was reduced but the ∆*Momcp1* mutant showed no significant change under the Mn^2+^ treatment (Figure 5B). Therefore, the different changes of various metals between the WT and ∆*Momcp1* mutant indicated that the blast fungus could modulate the contents of different metals under Mn^2+^ stress.

### 2.6. Differential Gene Expression Analysis between ∆Momcp1 Mutant and WT Strains Using Transcriptomes

In order to identify the differently expressed genes (DEGs) in ∆*Momcp1* compared with WT during hypha growth, transcriptomes were performed. According to the results, 1300 genes were up-regulated and 1206 genes were down-regulated in ∆*Momcp1* compared with WT (Appendix A). The Gene ontology (GO) enrichment and Kyoto Encyclopedia of Genes and Genomes (KEGG) pathway analysis were used for further analysis (Appendix A). To explore the compromised pathogenicity in the ∆*Momcp1* mutant, we determined many pathogenicity related genes in our RNA-seq results, and there are many genes regulated in the ∆*Momcp1* mutant (Figure 6, Table 1) [23,24,25,26,27,28,29,30,31,32,33,34,35,36,37,38,39,40,41,42,43,44,45,46]. Interestingly, several surface and signal recognition genes showed much higher or lower expression levels than the other pathogenicity related genes, indicating that the expressions of these genes were regulated in the ∆*Momcp1* mutant.

### 2.7. Transcriptome Analysis of the Sensitivity of Mn^2+^ in △Momcp1 Mutant

Because of the sensitivity of ∆*Momcp1* to Mn^2+^, we analyzed the DEGs in WT and ∆*Momcp1* mutant strains under a Mn^2+^ treatment. 2306 DEGs responded to excess Mn^2+^ and were shared in both the WT and ∆*Momcp1* mutant. There were 797 and 1581 Mn related DEGs particularly found in the WT and ∆*Momcp1* mutant, respectively (Appendix A). According to the GO enrichment, the genes relating to the purine nucleoside triphosphate metabolic process, macromolecular complex and structure constituent of ribosome were down-regulated in the ∆*Momcp1* mutant (Appendix A). However, the genes involving the cellular protein catabolic process, proteasome core complex and threonine-type peptidase activity showed less expression in the WT (Appendix A). Meanwhile, there was no significant enrichment of up-regulated genes both in the ∆*Momcp1* mutant and WT (Appendix A). From the KEGG analysis, the expressions of proteasome and oxidative phosphorylation related genes were significantly down-regulated in the WT and ∆*Momcp1* mutant, respectively (Figure 7A,C). As for the up-regulated gene enrichment, the pathways involving the pyrimidine metabolism, peroxisome, DNA replication and purine metabolism were enriched in the ∆*Momcp1* mutant (Figure 7D). Moreover, only the peroxisome pathway was found in the wild type (Figure 7B).

We also analyzed the DEGs involved in organelle proteins, including the mitochondrion and vacuole. Besides the shared genes found both in the WT and ∆*Momcp1* mutant, there were many specific genes responding to Mn^2+^. Furthermore, the numbers of specific genes were much more in the ∆*Momcp1* mutant than in the WT. In terms of the mitochondrion related genes, there were 21 DEGs shared both in the WT and ∆*Momcp1* mutant. However, 18 specific genes were found in the ∆*Momcp1* mutant (Appendix A). Similarly, the specific genes encoding vacuolar proteins were only found in the ∆*Momcp1* mutant, besides 9 shared DEGs (Appendix A). We constructed the protein-protein interaction network relating to organelle DEGs in the WT and ∆*Momcp1* mutant. Interestingly, much more complicated interaction networks and higher numbers of nodes were generated in the ∆*Momcp1* mutant (Appendix A). 

### 2.8. Overexpression of MnSOD Restored the Mn^2+^ Tolerance in ∆Momcp1 Mutant

According to transcriptome and antioxidant enzyme assays, we performed an overexpression of *MnSOD* in the ∆*Momcp1* mutant. Under a higher concentration of MnCl_2_, the growth of the ∆*Momcp1* mutant was inhibited completely, but the strain with an overexpression of *MnSOD* was more tolerant (Figure 8A,B). Similarly, the biomass of ∆*Momcp1* was less than that of the overexpressed strain (Figure 8A). Therefore, the overexpression of *MnSOD* restored the tolerance in the ∆*Momcp1* mutant under excessive Mn^2+^.

## 3. Discussion

With the development of the industrialized and urbanized world, heavy metal contamination is an increasing problem which raises health risks because of its accumulation in the food chain. Thus, exploring the inhibitory mechanism around the excessive accumulation of heavy metals and the elimination in organisms is very necessary. Mn^2+^ plays an essential role in diverse cellular processes, especially in the activities of many enzymes. Meanwhile, the excess amount of Mn^2+^ threatens the normal physiological function. There were many reported genes that involved the detoxification of Mn^2+^ that were found in many organisms, and most were transporting related genes and Mn^2+^ dependent enzymes. Here, we identified a cytochrome P450 gene, *MoMCP1* participating in the Mn^2+^ metabolism in *M. oryzae*. The ∆*Momcp1* mutant showed more sensitivity to Mn^2+^, as well as decreased pathogenicity during infection. In order to explore the mechanism in the ∆*Momcp1* mutant, transcriptomes were performed, and the results implied that the expressions of mitochondria, vacuole, and energy synthesis associated genes were regulated, making the fungus sensitive to Mn^2+^ in the ∆*Momcp1* mutant.

### 3.1. Pathogenicity Related Genes

The ∆*Momcp1* mutant showed pathogenicity attenuation, and we thus compared the DEGs involved in pathogenicity between the ∆*Momcp1* mutant and WT (Table 1). Among these pathogenicity related DEGs, many signal recognition genes were induced with much higher or lower levels, such as Mpg1, Pth11, Wish and Mhp1. Mpg1 and Mhp1 are both hydrophobin proteins [23,46], whereas Pth11 and Wish, being G-protein coupled receptors, sense the hydrophobic components on the host surface [45,47]; these four genes are all involved in the process of infection-related morphogenesis, implying that *MoMCP1* might modulate these pathogenicity genes during infection. We expressed *MoMCP1* under a strong promoter and found no appressorium formation, implying the relationship between *MoMCP1* and appressorium formation. Furthermore, the Mn^2+^ toxicity decreased the pathogenicity with the punch inoculation, and the ∆*Momcp1* mutant also showed a lower pathogenicity than the WT under the Mn^2+^ treatment. In reference to transcriptome results, many pathogenicity related genes were also regulated by Mn^2+^, indicating that Mn^2+^ plays important roles in fungal development and infection (Appendix A) [48,49,50,51,52,53,54,55,56,57].

### 3.2. Mitochondria Related Genes

Mitochondria, as the power production industry, provide a host of metabolic functions through oxidative phosphorylation. Evidences about the manganese neurotoxicity causing mangaism and Parkinson’s disease have been reported, and the common cellular mechanisms are excessive Mn^2+^ accumulated within mitochondria and oxidative stress induced preferentially [58]. According to the KEGG pathway, we found that the mitochondria and oxidative phosphorylation related genes were induced in the ∆*Momcp1* mutant under excessive Mn^2+^ (Appendix A). In yeast, the manganese trafficking factor for mitochondria, Mtm1, is responsible for importing Mn^2+^ into mitochondria to provide for SOD2 synthesis [59]. From our results, many protein import machineries of mitochondria-related DEGs were found to be involved in the formations of translocase of the outer mitochondrial membrane (TOM) or translocase of the inner mitochondrial membrane (TIM) complexes. In particular, *Tom40*, the channel-forming subunit, and *Tom22*, the central receptor, were up-regulated in the ∆*Momcp1* mutant. However, the TOM complex is the main gate for mitochondrial protein entry and homeostasis [60,61], whether the TOM complex responded to Mn^2+^ or Mn^2+^ binding proteins is unknown. Furthermore, the gene encoding the mitochondrial DNA replication protein was down-regulated in the ∆*Momcp1* mutant, and the dynamics of mitochondrial DNA is implicated in human diseases [62]. Meanwhile, excessive Mn^2+^ decreases the expressions of oxidative phosphorylation related genes involved in five constituted complexes in the ∆*Momcp1* mutant. Oxidative phosphorylation is represented in the adenosine triphosphate (ATP) production, and many genes encoding ATP synthase subunits also showed decreased expressions, implying that excessive Mn^2+^ might reduce the energy production and fungal biomass in the ∆*Momcp1* mutant. In Δ*mntP Escherichia coli*, a highly manganese-sensitive strain, manganese stress affected energy metabolism pathways including oxidative phosphorylation and ATP synthesis [63]. There are many fungicides targeting for the electron respiration chain, such as strobilurins, but the target site is limited, which may generate the resistance to these fungicides in pathogens [64]. Here, we found that Mn^2+^ decreased the expression of multiple target genes and might cause mitochondrial dysfunction and disrupted the energy production in the ∆*Momcp1* mutant.

### 3.3. Vacuole Related Genes

Vacuoles are essential for fungal growth with diverse cellular functions involved in the storage and degradation of components and nutrition, as well as the maintenance and regulation of homeostasis and transport. For excess Mn^2+^, the expressions of almost vacuolar associated genes were down-regulated, especially vacuolar protein sorting-associated genes (VPS), which are involved in the sorting and transport of proteins to the vacuoles, according to the transcriptome analysis (Appendix A). Many VPS proteins participate in retromer complex formation. For example, Vps35, Vps29 and Vps26 participate in cargo selection, and the dimer of Vps5 and Vps17 participates in tubule or vesicle formation [65]. In *S. cerevisiae*, many mutants with VPS related gene deletion showed sensitivity to Cd [66]. Besides the shared VPS genes in the WT and ∆*Momcp1* mutant, VPS9, VPS17, VPS26, VPS35, and VPS45 were down-regulated specifically in ∆*Momcp1*. In *Cryptococcus neoformans*, VPS45 can mediate the trafficking for iron uptake, mitochondrial function and virulence [67]. In *M. oryzae*, MoVPS35, MoVPS26 and MoVPS29 constituted a cargo-recognition complex of the retromer, and MoVPS35 is responsible for autophagy based membrane trafficking events [68]. Similar functions of FgVps35 and FgVps26 were also found in *Fusarium graminearum* [69]. Interestingly, Sarkar et al. (2019) found that Mn exposure reduced VPS35 in lipopolysaccharide (LPS) primed microglial cells, which promote the degradation of Mfn2 (Mitofusin 2) and increase mitochondrial dysfunction [70]. Their results provide the association between mitochondrial and vacuolar genes under excessive Mn^2+^. Furthermore, MoVPS17 is a sorting nexin and localized to endosomes, which is essential for fungal development and infection [38]. *MoVps9* is involved in autophagy and endocytosis [71]. The vacuole fusion gene, *MoMon1* is down-regulated under Mn^2+^ toxicity in the ∆*Momcp1* mutant and is required for vacuolar assembly, autophagy, fungal development and pathogenicity in *M. oryzae* [35]. Likewise, *FgMon1*, a homolog gene in *F. graminearum*, is also important for vacuole fusion, autophagy and plant infection [72]. Therefore, Mn^2+^ toxicity might disrupt the expressions of normal routes of cargo transport and vacuole integrity related genes in the ∆*Momcp1* mutant.

### 3.4. Peroxisome Related Genes, Especially MnSOD, Detoxify Mn^2+^ Induced Oxidative Stress

In eukaryotic organisms, peroxisomes play important roles in several metabolism processes, such as ROS elimination, fatty acids β-oxidation, glyoxylate cycle and secondary metabolite biosynthesis. When cells are exposed by UV light and different oxidizing agents, peroxisome proliferation with the formation of tubular peroxisome and up-regulation of peroxins (PEX) related genes will be induced [73]. In *M. oryzae*, many peroxisome related genes are identified and influence pathogenicity [74]. We found that many up-regulated DEGs involving peroxisomes were enriched both in the WT and ∆*Momcp1* mutant (Appendix A). In *C. neoformans*, PEX1 and PEX6 encoding AAA-type proteins are essential for peroxisome biogenesis and fatty acid utility [75]. Furthermore, *MoPex6* participated in the peroxisome mediated β-oxidation of long-chain fatty acids in *M. oryzae*. PEX16 functions as a peroxisome and Woronin body formation in *Aspergillus luchuensis* [76]. Therefore, peroxisome formation and fatty acid metabolism might be strategies for excessive Mn^2+^ detoxification. However, excess Mn^2+^ is regarded as enhancing the production of ROS [77], and peroxisomes are also responsible for ROS detoxification. However, there are many enzymes, such as catalase, peroxidase, glutathione S-transferases, and peroxiredoxins, which involve ROS detoxification and are abundant in peroxisomes [74]. We found that expressions of these ROS detoxification related genes were down-regulated both in the WT and ∆*Momcp1* mutant, while there was only one up-regulated gene, glutathione S-transferase II, in the WT. Moreover, SOD related enzymes are used to clean up the superoxide anion, which is more harmful to the cell than hydrogen peroxide. SOD was reported to localize in the mitochondria, but it was found in rat liver peroxisomal membranes [78], which was consistent with the KEGG analysis. On comparing the expression difference relating to superoxide dismutase related genes, the expression of *SOD2* (*MnSOD*) in the WT was up-regulated but the expression of this gene in the ∆*Momcp1* mutant was reduced. In line with the activities of antioxidant enzymes, SOD enzymes might play a dominant role under Mn^2+^ stress, instead of peroxidase and catalase. Furthermore, we overexpressed a *MnSOD* in the sensitive mutant with less inhibitory rates under Mn toxicity (Figure 8), implying that a higher expression of *SOD2* could provide more Mn binding proteins [79] and decrease oxidative damage. Meanwhile, excessive Mn^2+^ increased H_2_O_2_ production and led to cell death [79], which is consistent with our results (Figure 4A), but the activities of enzymes that catalyzed H_2_O_2_ were not significantly different, indicating that there might be some non-enzyme compounds responsible for H_2_O_2_ degradation. Overall, the up-regulated expression of *MnSOD* is a considerable reason for the excess Mn^2+^ detoxification in the WT.

### 3.5. Proteasome Related Genes

The ubiquitin-proteasome pathway as a primary cytosolic proteolytic machinery plays important roles in the selective degradation of various forms of impaired proteins. The expressions of proteasome related genes were decreased in the WT according to the KEGG analysis (Appendix A). The 26S proteasome is composed of two multi-subunit complexes. One is a 20S proteasome named as a catalytic core, while the other is a 19S proteasome named as a regulatory particle. Under heavy metal stress, proteasomes should have been responsible for the degradation of damaged proteins, but it has been reported that heavy metal decreased the activities of the proteasomes [80]. However, the down-regulation of the proteasome pathway was only enriched in the WT, not the Mn^2+^ sensitive mutant, indicating that the regulation of the proteasome might be involved in Mn^2+^ tolerance. In *Arabidopsis*, a component of the 26S proteasome complex, ARS5 negatively regulates thiol biosynthesis and arsenic tolerance [81]. Furthermore, alternative formations of 20S proteasomes isolates through the decreased formation of α3 and increased α4-α4 proteasome levels confer the resistance to heavy metals in yeast and human cells [82,83]. We also found that the expression of the α3 subunit was reduced, but that no changed expression of the α4 subunit was observed. Therefore, the associations between the Mn^2+^ resistance and various proteasome isolates need further researches.

### 3.6. Nucleotide Synthesis Related Genes

There were 3 pathways, including the purine metabolism, pyrimidine metabolism, and DNA replicate, specifically enriched in the ∆*Momcp1* mutant (Appendix A). Purine and pyrimidine provided the materials for nucleoside syntheses, and there are many DNA and RNA synthesis related genes with up-regulation. Furthermore, the genes that participated in the DNA replicate were mainly involved in the DNA repair. For example, a MSH2 homolog gene was up-regulated in the ∆*Momcp1* mutant, which could form different heterodimers with MSH6 or MSH3 to recognize the base mismatches or large insertion-deletion loops [84]. Meanwhile, we also found a down-regulated gene, *MoNim1*, that is involved in the DNA damage checkpoint regulator and appressorium mediated plant infection [85]. In the ∆*Momcp1* mutant, a higher amount of H_2_O_2_ accumulation induced by excessive Mn^2+^ caused DNA damage; additionally, the up-regulation of the purine metabolism, pyrimidine metabolism, and DNA replicate might be the compensatory methods in the ∆*Momcp1* mutant.

### 3.7. Secreted Compound and Transport Related Genes

Besides the pathways affected by excessive Mn^2+^, there are many molecular mechanisms in fungi under heavy metal stress, including secreted metal binding compounds, the cell wall, transport and damage alleviation [86]. Therefore, we search the expressions of these related genes from our transcriptomes, in combination with previous data [87]. We analyzed the expressions of genes involved in fatty acid oxidation, the phosphate and carboxylate metabolism pathway, carbohydrate degradation, and transporters for sugar, organic acid and drugs (Appendix A). Many DEGs were found both in the WT and ∆*Momcp1* mutant after the Mn^2+^ treatment. The numbers of fatty acid oxidation related genes in the ∆*Momcp1* mutant were more than for the WT, indicating that *MoMCP1* might participate in the fatty acid oxidation pathway for Mn^2+^ detoxification (Appendix A). Transporter related genes play critical roles against heavy metal and xenobiotic compounds, there are many transporters regulated excess Mn^2+^ that have been reported [17,88]. Interestingly, the expressions of almost sugar transporter related genes were down-regulated in WT, but the expressions of these genes were up-regulated in the mutant (Appendix A). In *S. aureus*, ArlRS regulated the needs of amino acids and sugars to cope with calprotectin induced manganese starvation [18]. Furthermore, many genes encoding secreted carbohydrate degradation enzymes were regulated, such as chitinase, chitin deacetylase, and endoglucanase, these genes were recorded in fungi responding to heavy metals [89,90]. But how these enzymes chelate excessive Mn^2+^ is unknown. Therefore, further metabolism analyses are needed to determine the main substances involved in Mn^2+^ detoxification.

### 3.8. Changes of Other Metals by Excessive Mn^2+^

To face excessive Mn^2+^ stress, we found that the contents of Ca, Zn, and Cu increased both in the WT and ∆*Momcp1* mutant. However, excessive Mn^2+^ can inactivate many metal-binding proteins through replacement. According to the Irving Williams series [91], increasing contents of Zn and Cu might be involved in detoxification through the correction of Mn^2+^ binding proteins and recover activities. Ca as a cellular signal plays an essential role in blast fungal development and pathogenicity [92]. Recent results provided evidence for a Ca^2+^ dependent regulator, MICU1, against Mn^2+^ toxicity through the modulation of a mitochondrial Ca^2+^ uniporter [93]. We also identified a calcium-binding mitochondrial carrier protein, which might increase the Ca content to decrease Mn^2+^ toxicity. Furthermore, the Fe content in the WT was decreased, but there was no variation in the ∆*Momcp1* mutant. Many results relating to the interdependency of transport and regulation between Mn and Fe have been found, and Mn can out-compete Fe for these common Fe-binding sites. Additionally, an increased content of Mn is associated with a reduced amount of Fe [94]. The accumulation of excessive Fe causes various harmful ROS for the cell through the Fenton reaction [95]. We found that Fe^3+^ associated genes had a higher expression in the WT. For example, siderophore production is induced by iron deficiency [96], which is consistent with the lower content of iron in the WT. Moreover, the deletion of the siderophores production gene leads to sensitivity to H_2_O_2_ in *Alternaria alternate* [97]. Thus, the hemostasis of different metals might provide a solution for Mn^2+^ stress.

### 3.9. Cytochrome P450 Genes and Mn Stress

In the fungal kingdom, the diversity of Cytochrome P450 (CYP) helps fungal primary and secondary metabolite synthesis and the adaption to various ecological niches, especially xenobiotic degradation. In *M. oryzae*, *MoCYP51a* is essential for conidiogenesis and fungicide tolerance [30]. Similarly, fungal development and stress responses were also disrupted in *Fusarium graminearum* with P450 gene mutants [98]. For metal responses, many results related to CYP450 genes have been established [99,100,101]. Furthermore, the polymorphism of the CYP450 gene, *CYP2D6L*, might have a relationship with manganese-induced neurotoxicity in chronic manganism patients [102].

According to our transcriptome analysis, the expressions of many genes were regulated in the ∆*Momcp1* mutant, but whether there are direct roles between *MoMCP1* and Mn^2+^ needs to be proved in the future. Furthermore, we used a transcriptomics analysis to find the DEGs, but whether these DEGs are really responsible for targeted pathways or organelles requires other evidence such as an ultrastructure observation, along with physiological and metabolic experiments. Therefore, our results provide more information about Mn regulating network in rice blast fungi (Figure 9) and references for Mn induced diseases in humans.

## 4. Materials and Methods

### 4.1. Strains and Growth Conditions

The *M. oryzae* YN125 strain was used as a wild type in this study. All strains were grown on PSA (2% potato, 1% sucrose and 1.5% agar) plates for 5–10 days in the dark at 28 °C. The conidia were harvested from oats meal agar (OTA) plates (4% oats meal, 1.5% agar) after a 10-day-old inoculation with a 12 h photoperiod. The solid complete medium (CM: 0.6% yeast extract, 0.6% casein hydrolysate, 1% sucrose, 1.5% agar) was used for mycelial growth. For the transformation selection, hygromycin (Roche, Indianapolis, IN, USA) was added to the final concentrations of 300 μg mL^−1^ in TB3 (20% sucrose, 0.3% yeast extract, 0.3% casein hydrolysate, and 1.5% agar) plates.

### 4.2. Mycelial Growth, Mycelial Dry Weight, Conidial Production, Germination, Appressorium Formation and Plant Infection Assays

The colony diameters were measured in CM plates after a 7-day inoculation. For the mycelial dry weight measurement, mycelium was obtained from 2-day-old inoculation from liquid CM with the same amount of initial hypha and dried until the weight did not change. For the conidial production, the spores were washed from the OTA culture, and added with Mn^2+^ for 10 days by 5 mL distilled water, after which the numbers were counted under a microscope.

For the pathogenicity assessment, the 5 mL conidial suspension (1 × 10^5^ conidia mL^−1^ in 0.2% gelatin) was sprayed on 4-week-old susceptible rice seedlings Lijiangxintuanheigu (LTH). The inoculated plants were placed in a moist chamber at 28 °C for the first 24 h in darkness, and then transferred to a growth chamber. The disease index was determined at 7 days after inoculation according to a previously described method [103]. For the punch inoculation, the detached rice leaves were lightly wounded with a mouse ear punch, the 10 μL conidia suspensions treated with Mn^2+^ or not treated were dropped on the wound. The lesions were photographed at 7 days after inoculation. The fungal biomass in the infected rice leaf tissue was quantified by a previously described method [104].

### 4.3. Gene Deletion

The *M. oryzae* protoplast preparation and fungal transformation were performed following standard protocols [105]. The replacement vector pCX62 encoding hygromycin phosphotransferase (*HPH*) with a 798 bp upstream and 957 bp downstream flanking sequence fragment of the target gene was constructed. The primers used to amplify the flanking sequence are listed in Appendix A. The knock out candidates were identified by PCR with target gene specific primers. We also designed two primer pairs for further confirmation, one primer located inside the *HPH* gene and the other one located outside the upstream or downstream homology arm, respectively. The amplified fragments from a transformant were cloned into T19 vector (Takara) and sequenced to ensure the donor gene was inserted into the right location.

### 4.4. RNA Isolation and RT-qPCR

Fungal mycelium from liquid CM was used to extract fungal RNA for 2 days at 28 °C in a 150 rpm shaker. The rice leaves for the RNA extraction were collected at 0, 24, 48, and 72 h after inoculation. The total fungal and rice RNA samples were extracted by RNeasy Plus Mini Kit (Qigen, Hilden, MA, USA). The first strand cDNA was synthesized with Transcriptor First Strand cDNA Synthesis Kit (Roche, Indianapolis, IN, USA). For a quantitative real-time PCR, the primers used for target genes were listed in S13. The quantitative real-time PCR was performed with Bio-rad using *SYBR* Premix *Ex Taq* (Takara, Kusatsu, Shiga, Japan). The relative quantification of the transcripts was calculated by the method [106].

### 4.5. RNAseq Analysis

The wild type and mutant strains were grown in liquid CM for two days at 28 °C in a 150 rpm shaker. Following this, the mycelia from each strain were transferred to a new liquid CM and incubated at 28 °C in a 150 rpm shaker. The liquid CM was added with 2 mM Mn^2+^ as treatment. After a two-day inoculation, the mycelia were filtered and prepared for RNA extraction. These experiments were performed with three biological repeats. The libraries were sequenced on the Illumina HiSeqTM 4000 platform by Novogene Co. (Beijing, China). The genome of *Magnaporthe oryzae* 70-15 was used as the reference genome and downloaded at NCBI (https://www.ncbi.nlm.nih.gov/genome/62?genome_assembly_id=22733, accessed on: 31 March 2016). The differential expression genes were analyzed by DESeq2 R package and selected based on a false discovery rate (FDR) ≤0.05 between treatment and control within a strain. Gene Ontology (GO) enrichment and KEGG pathway analysis of differentially expressed genes were implemented by the cluster Profiler R package. The transcriptome datasets can be retrieved from the NCBI SRA database under Project ID PRJNA523930.

### 4.6. Generation of Transformants Expressing the MoMCP1–eGFP

The *MoMCP1* coding sequence was amplified from cDNA and inserted into pDL2 using the yeast gap repair approach. The fusion constructs were confirmed by sequencing and transformed into YN125. The hygromycin B resistant transformants were selected and confirmed by PCR and fluorescent observation.

### 4.7. Overexpression of MnSOD in ∆Momcp1

The *MnSOD* coding sequence was amplified from DNA and inserted into pYF11 with a bleomycin resistant gene using the yeast gap repair approach, before being transformed into ∆*Momcp1*. The bleomycin (200 μg/mL, Invitrogen, Eugene, OR, USA) resistant transformants were selected and confirmed by PCR and fluorescent observation.

### 4.8. Determination of H_2_O_2_ Content, Activities of SOD, CAT and POD

The mycelia were harvested from liquid CM after a two-day inoculation, and transferred into new liquid CM added with 2 mM Mn^2+^. The two-day inoculation mycelia were frozen by liquid nitrogen. The H_2_O_2_ content, activities of SOD, CAT and POD, were determined according to the technical bulletins about fluorometric hydrogen peroxide assay kit (Sigma, St. Louis, MO, USA), SOD assay kit (Sigma, St. Louis, MO, USA), catalase assay kit (Sigma, St. Louis, MO, USA), and peroxidase assay kit (Sigma, St. Louis, MO, USA). The fungal biomass was determined by the protein content according to previous protocols [107]. These experiments were performed three independent times.

### 4.9. Determination of Different Metal Contents by ICP-OES

The cultural methods used on the materials were the same as for antioxidant enzyme determination, and the materials were dried until there were changes of weight. 0.2 g of dried samples were digested in a microwave oven at 200 °C using 5 mL nitric acid and 2 mL perchloric acid mixed solutions. The digested solutions were transferred into 100 mL volumetric flasks and the contents of the metals were determined by ICP-OES (Optima8300, PerkinElmer cooperation, Waltham, MA, USA).

## Figures and Tables

**Figure 1 ijms-20-01590-f001:**
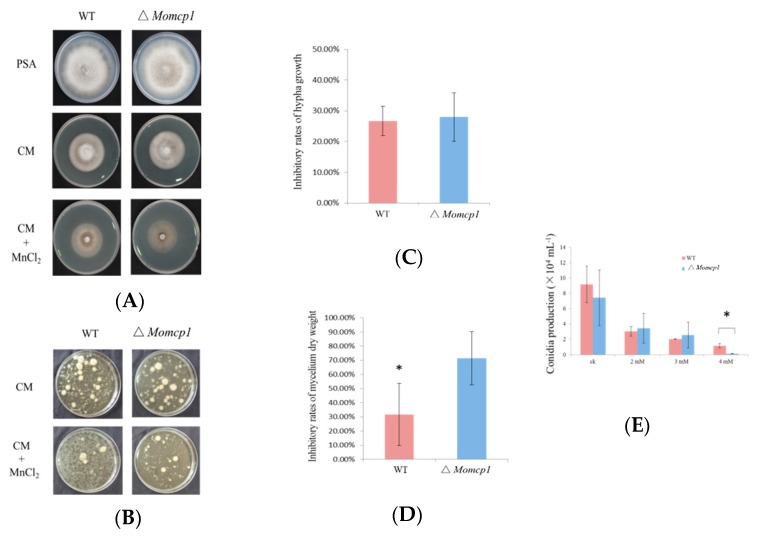
∆*Momcp1* is sensitive to Mn^2+^. (**A**) The appearances of colonies under different plates. (**B**) The mycelium obtained from liquid CM for two days with or without 2 mM Mn^2+^ treatment. (**C**) The inhibitory rates of hypha growth under a 6 mM Mn^2+^ treatment. The colony diameter of each strain was measured at seven days after inoculation at CM plates containing 6mM Mn^2+^. (**D**) The inhibitory rates of the mycelium dry weight under a 2 mM Mn^2+^ treatment. The dry weight of mycelium of each strain was measured at two days after inoculation into the Mn^2+^ containing liquid CM. (**E**) The conidia production. The numbers of conidia were counted at ten days after inoculation on Mn^2+^ containing OMA plates. The significant difference was determined by a *t*-test; *, *p* < 0.05.

**Figure 2 ijms-20-01590-f002:**
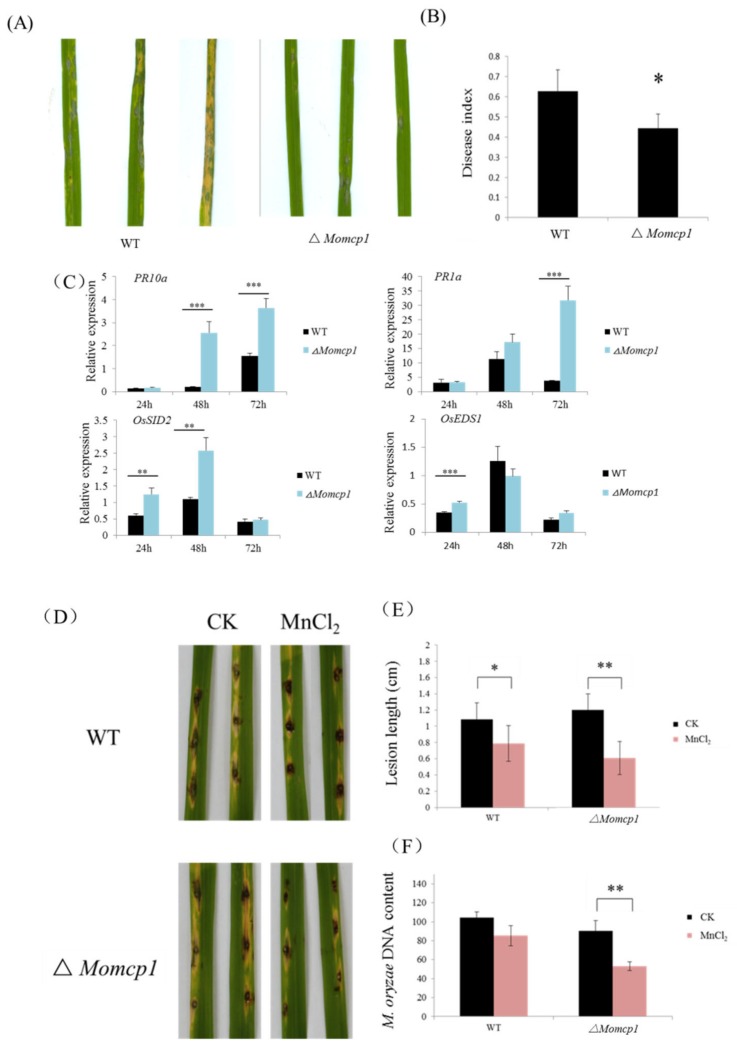
∆*Momcp1* mutant showed a compromised pathogenicity to rice. (**A**) The lesions on Lijingxintuanheigu (LTH) seedlings infected by different strains. The photographs were taken at 7 dpi (day post inoculation). (**B**) The disease indexes were obtained from 3 times and more than 30 seedlings were counted each time. The significant difference was determined by a *t*-test; *, *p* < 0.05. (**C**) The expression levels of the defense related genes at 24, 48, and 72 hpi after each strain inoculation on LTH. The experiments were repeated three times. The significant differences determined by a Tukey’s honestly significant difference (Tukey-HSD) test; **, *p* < 0.01; ***, *p* < 0.001. (**D**) Punch inoculation of each strain on the detached leaves. The detached leaves were wound by ear needle, and the conidia suspensions added or not added with 2mM Mn^2+^ were dropped on the wound. The pictures were taken at 7 dpi. (**E**) The length of the lesion was measured at 7 dpi. (**F**) Fungal biomass was calculated at 7 dpi. The significant difference was determined by a *t*-test; *, *p* < 0.05; **, *p* < 0.01.

**Figure 3 ijms-20-01590-f003:**
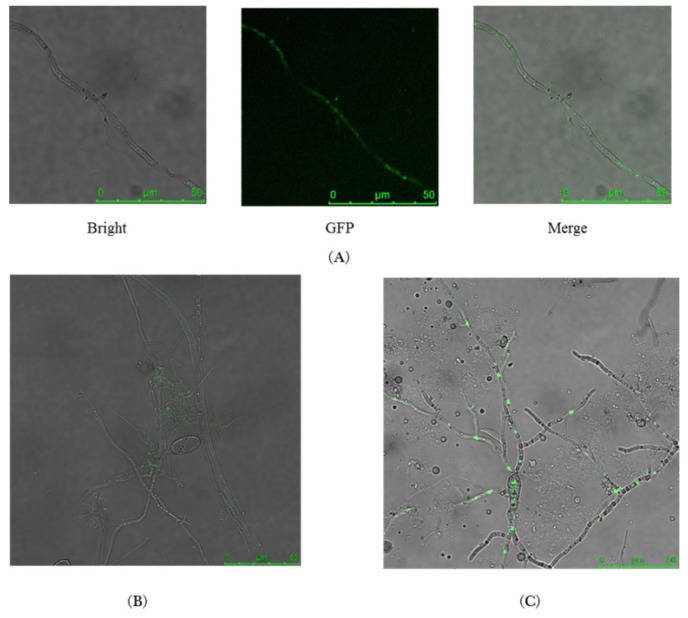
Detection of MoMCP1-GFP in *M. oryzae*. (**A**) The distribution of MoMCP1-GFP in the hypha, and (**B**) conidia; (**C**) The distribution of the green signal in the branched hypha that germinated from the conidia; scale bar, 50 µm.

**Figure 4 ijms-20-01590-f004:**
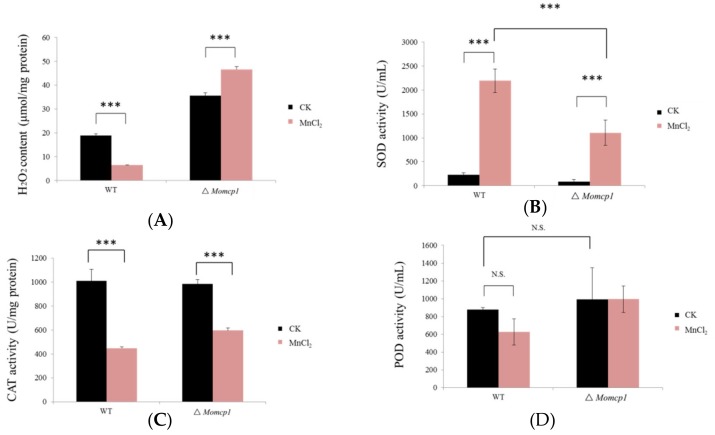
The changes of H_2_O_2_ contents and activities of antioxidant enzymes in the WT and ∆*Momcp1* mutant under 2 mM Mn^2+^ treatment. (**A**) H_2_O_2_ contents. (**B**) SOD activity. (**C**) CAT activity. (**D**) POD activity. Each bar represents the means ± SD of three independent experiments. The significant differences were determined by Tukey-HSD test; ***, *p* < 0.001; N.S, no significance.

**Figure 5 ijms-20-01590-f005:**
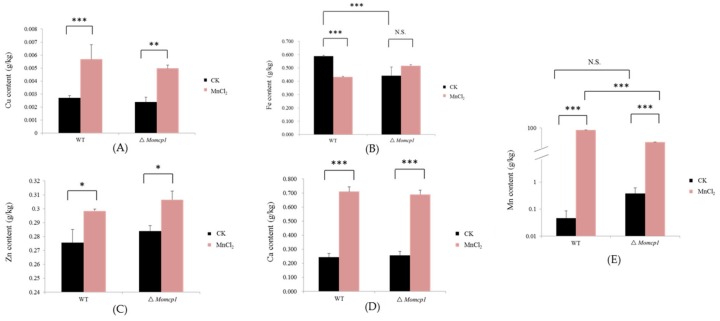
The contents of different metals in the WT and ∆*Momcp1* mutant under 2 mM Mn^2+^. The contents of Cu (**A**), Fe (**B**), Zn (**C**), Ca (**D**), and Mn (**E**) in the WT and ∆*Momcp1* mutant. The significant differences determined by the Tukey-HSD test; *, *p* < 0.05; **, *p* < 0.01; ***, *p* < 0.001; N.S, no significance.

**Figure 6 ijms-20-01590-f006:**
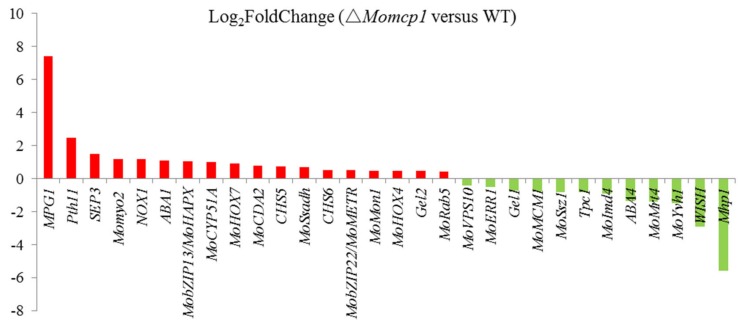
The different expressions of pathogenicity genes in *Mocmp1* mutants compared with WT.

**Figure 7 ijms-20-01590-f007:**
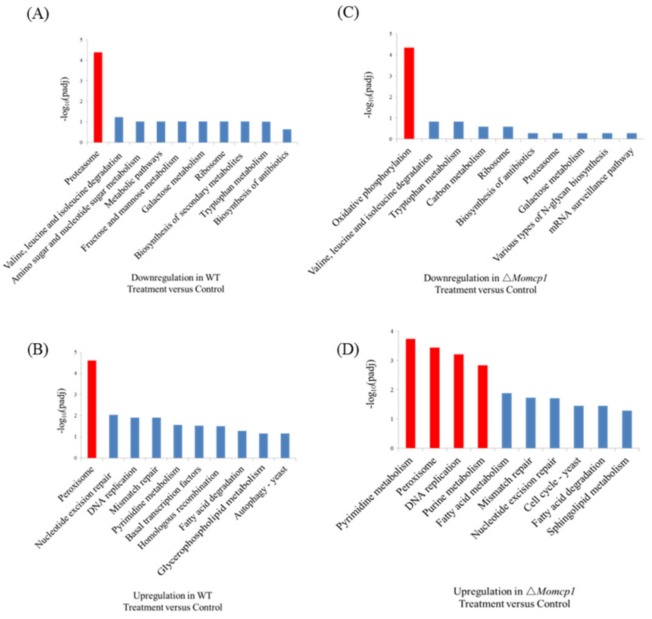
The KEGG pathways in WT and ∆*Momcp1*. (**A**) Down-regulation and (**B**) up-regulation of pathways in WT under 2mM Mn^2+^; (**C**) Down-regulation and (**D**) up-regulation of pathways in ∆*Momcp1* under 2mM Mn^2+^.

**Figure 8 ijms-20-01590-f008:**
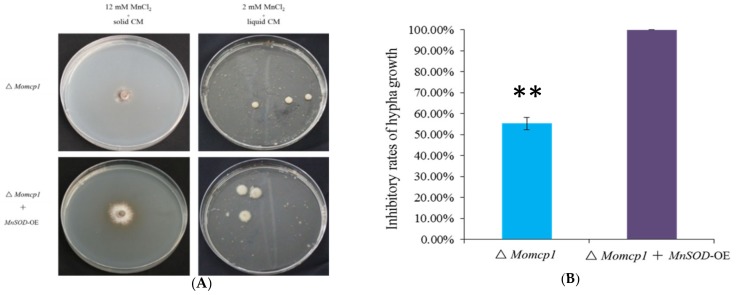
Overexpression of *MnSOD* restored the Mn^2+^ tolerance in ∆*Momcp1* mutant. (**A**) The hypha growth of ∆*Momcp1* and ∆*Momcp1* with the overexpression of *MnSOD* under excessive Mn^2+^ (Left panel, 12 mM Mn^2+^ in solid CM. Right panel, 2 mM Mn^2+^ in liquid CM). (**B**) The inhibitory rates of the hypha growth in solid CM with excessive MnCl_2_. The significant difference was determined by a *t*-test; **, *p* < 0.01.

**Figure 9 ijms-20-01590-f009:**
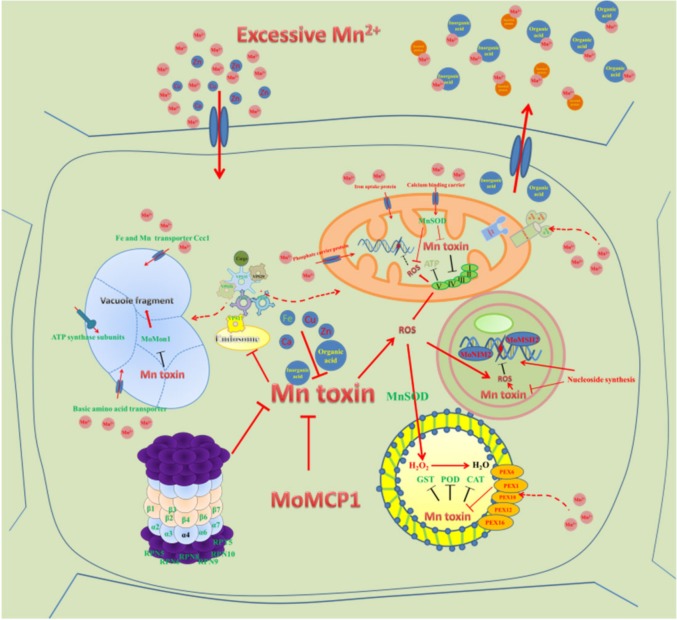
The putative gene network about *MoMCP1* alleviating Mn^2+^ toxin in *M. oryzae*. The colored arrows mean the induced or promoted reactions. The dotted arrows mean the putative associations. The “T” bars mean the inhibitory reactions.

**Table 1 ijms-20-01590-t001:** The differently expressed pathogenicity related genes in ∆*Momcp1*.

Gene ID	Description	Log_2_FoldChange (△*Momcp1* versus Wild Type)	Name	Reference
MGG_09134	hypothetical protein	7.42	*MPG1*	[23]
MGG_05871	hypothetical protein	2.45	*Pth11*	[24]
MGG_01521	cell division control protein 3	1.47	*SEP3*	[25]
MGG_03060	myosin type II heavy chain	1.16	*Momyo2*	[26]
MGG_00750	cytochrome b-245 heavychain subunit beta	1.16	*NOX1*	[27]
MGG_07626	cytochrome P450 monooxygenase	1.07	*ABA1*	[28]
MGG_05959	bZIP transcription factor	1.05	*MobZIP13/MoHAPX*	[29]
MGG_04628	cytochrome P450 51	1.01	*MoCYP51A*	[30]
MGG_12865	hypothetical protein	0.91	*MoHOX7*	[31]
MGG_08774	chitin deacetylase	0.78	*MoCDA2*	[32]
MGG_13014	class V chitin synthase	0.75	*CHS5*	[33]
MGG_01230	succinate-semialdehyde dehydrogenase	0.69	*MoSsadh*	[34]
MGG_13013	chitin synthase 8	0.51	*CHS6*	[33]
MGG_14561	regulatory protein Cys-3	0.49	*MobZIP22/MoMETR*	[29]
MGG_05755	vacuolar fusion protein	0.48	*MoMon1*	[35]
MGG_06285	hypothetical protein	0.46	*MoHOX4*	[31]
MGG_06722	1%2C3-beta-glucanosyltransferase	0.46	*Gel2*	[36]
MGG_01185	GTP-binding protein ypt5	0.40	*MoRab5*	[37]
MGG_00506	hypothetical protein	−0.42	*MoVPS10*	[38]
MGG_16126	hypothetical protein	−0.51	*MoERR1*	[39]
MGG_07331	1%2C3-beta-glucanosyltransferase	−0.72	*Gel1*	[36]
MGG_02773	MADS box protein	−0.77	*MoMCM1*	[40]
MGG_02842	hsp70-like protein	−0.81	*MoSsz1*	[41]
MGG_01285	C6 finger domain-containing protein	−0.82	*Tpc1*	[42]
MGG_03699	inosine-5’-monophosphate dehydrogenase	−0.91	*MoImd4*	[43]
MGG_07514	3-oxoacyl-[acyl-carrier-protein] reductase	−1.34	*ABA4*	[28]
MGG_08908	mRNA turnover protein 4	−1.39	*MoMrt4*	[44]
MGG_09700	tyrosine-protein phosphatase	−1.49	*MoYvh1*	[44]
MGG_09022	hypothetical protein	−2.91	*WISH*	[45]
MGG_10105	hypothetical protein	−5.58	*Mhp1*	[46]

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
