# Peer review of "MoMCP1, a Cytochrome P450 Gene, Is Required for Alleviating Manganese Toxin Revealed by Transcriptomics Analysis in Magnaporthe oryzae"

_ijms, 2019, doi:10.3390/ijms20071590_

Round 1

Reviewer 1 Report

The manuscript reports the identification and the functional characterization of MoMCP1, a cytochrome  P450 from Magnaporthe oryzae.

The paper present several experiments, although the referee finds that the description of the experiments is not always clear and that conclusions are not always appropriate.

Some points to explain better this point:

1)      Paragraph 2.1, lane 98-99, can you really conclude that MoMCP1 participate in elimination Mn elimination. In Figure 1 the authors only talk about sensitivity to Mn.

2)      Line 116: with respect to the ability to suppress the expression…can this be said? Shouldn’t we conclude only that expression is higher? Also the paragraph is focussed on pathogenicity, but the paragraph talks also about defence genes. The two concepts should be linked better, as it stands it is not clear. The paragraph 3 also talks about pathogenicity, shouldn’t than all the experiments related to this aspect being grouped in only one paragraph?

3)      Conclusions in 2.5, again a strong conclusion, maybe better to use MAY BE THE REASON and IS the reason

4)      In the conclusions of 2.6: the blast fungi COULD modulate the content,….also why is it written that changes of Fe (which it is reported not changing) were more important?

5)      I suggest to substitute figure 7a with S3, and to write something in the text about these tables

6)      It is not clear why in paragraph 2.7 it is reported that 1300 genes were up-regulated and 1206 were down. In 2.8 797 and 1581 are reported to be regulated specifically in WT and mutants…please explain better in text

7)      Opposite to the results, the discussion is very detailed. Perhaps it could be organized in sub-paragraphs?

8)      Overall the English needs to be checked as often it is not clear what the authors are trying to say, few examples:

Line 32: manganese can cause toxin to cell.

Is it toxic? Or is the formation of toxin in the cell?

Line 35: Mn is also the materials for….???

Line  49: to decrease the toxin of excess…???

Line 72: in responded to blast….???

This to give few examples…

Minor points:

1)      What is the concentration of Mn in Fig.A,B,C and D?

2)      Paragraph 2.2, what is hpi, can it also be inserted in the abbreviation list?

3)      In Figure 2, what is dpi? can it also be inserted in the abbreviation list?

4)      Figure 4, is it possible to increase the contrast, it is difficult to see the GFP. Also the  reference bar is not visually clear

5)      Please define DEG

6)      Reports the year of publication when the citation is like: Corbin et al found…(year?)

Author Response

Response to Reviewer 1 Comments

Thanks for your kind and rapid comments, we all think it is valuable and necessary for us to improve our manuscript according to your suggestions. And the responses are following:

Point 1: Paragraph 2.1, lane 98-99, can you really conclude that MoMCP1 participate in elimination Mn elimination. In Figure 1 the authors only talk about sensitivity to Mn.

Response 1: Thanks for your kind suggestion, it is our fault to confuse the concepts and we revised this problem according to your comments. (See line 99-100 in red)

Point 2:  Line 116: with respect to the ability to suppress the expression…can this be said? Shouldn’t we conclude only that expression is higher? Also the paragraph is focussed on pathogenicity, but the paragraph talks also about defence genes. The two concepts should be linked better, as it stands it is not clear. The paragraph 3 also talks about pathogenicity, shouldn’t than all the experiments related to this aspect being grouped in only one paragraph?

Response 2: Thanks for your kind suggestion, we revised the incorrect description. (See line 116-117). As for two pathogenicity paragraphs, we wanted to explain the less pathogenicity of Momcp1 and Mn decreased the pathogenicity of Momcp1 significantly, separately. But we thought grouped in only one paragraph is pretty suitable according to your suggestion. (See line 116-122, fig 2)

Point 3: Conclusions in 2.5, again a strong conclusion, maybe better to use MAY BE THE REASON and IS the reason.

Response 3: Thanks for your kind suggestion, we revised the incorrect description. (See line 158-160)

Point 4:  In the conclusions of 2.6: the blast fungi COULD modulate the content,….also why is it written that changes of Fe (which it is reported not changing) were more important?

Response 4: Thanks for your kind suggestion, we regarded the WT as the tolerance strains, the changes of Fe in WT but no changes in mutant may indicate the Fe play an important role in Mn tolerance. But our poor description could cause ambiguous meaning for readers, and we revised this part. (See line 173-174)

Point 5:   I suggest to substitute figure 7a with S3, and to write something in the text about these tables

Response 5: Thanks for your kind suggestion, we have substitute figure 7a with S3. (See Fig 6 and table1)

Point 6:   It is not clear why in paragraph 2.7 it is reported that 1300 genes were up-regulated and 1206 were down. In 2.8 797 and 1581 are reported to be regulated specifically in WT and mutants…please explain better in text

Response 6: Thanks for your kind suggestion. The paragraph 2.7 (2.6 in new revision) want to compared the differently expression genes between WT and mutant without Mn treatment. Because it is hard to explain the less pathogenicity in mutant according to its phenotype, so the RNA-seq was performed. The 1300 and 1206 means the genes differently expressed in mutant. As for paragraph 2.8 (2.7 in new revision), 797 and 1581 means the unique Mn responding genes in WT and mutant, respectively. We have revised the incorrect sentences. (See line 183-185; line 208-209)

Point 7: Opposite to the results, the discussion is very detailed. Perhaps it could be organized in sub-paragraphs?

Response 7: Thanks for your kind suggestion. Due to huge data obtained from RNA-seq results, we want to find more and possible Mn target candidates, so the discussion parts are detailed. But we also need to testify that our results are credible and provide clues for other researchers focusing the Mn toxin, so the discussion seemed much more detailed. In order to make this part more orderliness, we added the titles at each part, but we did not know whether it is suitable. (See line 263, 277, 303, 328, 358, 374, 386, 405, and 423)

Point 8    Overall the English needs to be checked as often it is not clear what the authors are trying to say, few examples:

Line 32: manganese can cause toxin to cell.

Is it toxic? Or is the formation of toxin in the cell?

Line 35: Mn is also the materials for….???

Line  49: to decrease the toxin of excess…???

Line 72: in responded to blast….???

Response 8: Thanks for your kind suggestion. Actually these problems existed in our manuscript because of the poor English written. But we have tried our best to solve these problems. (See line 34, 37, 51, 74 and the sentences in red)

Point 9    What is the concentration of Mn in Fig.A,B,C and D?

Response 9: Thanks for your kind suggestion. It is our fault to neglect this important problem, and we have added the concentration of each experiment in the figure label. (See line 103-106; line 134; line 163; line 179; line 232-233)

Point 10-11    Paragraph 2.2, what is hpi, can it also be inserted in the abbreviation list?

 In Figure 2, what is dpi? can it also be inserted in the abbreviation list?

Response 10-11: Thanks for your kind suggestion. We checked and listed these unexplained abbreviation in the abbreviation list at the end of the manuscript. (See line 115 and 126, abbreviation list.)

Point 12    Figure 4, is it possible to increase the contrast, it is difficult to see the GFP. Also the  reference bar is not visually clear

Response 12: Thanks for your kind suggestion. We set the contrast again to make the GFP and reference bar clear, and added the explanation of reference bar in the figure label. (See line 144, figure 3)

Point 13 Please define DEG

Response 13: Thanks for your kind suggestion. The DEG means the differently expressed genes which are used in transcriptome analysis, and we add the explanation in the abbreviation list. (See line 183 and abbreviation list)

Point 14 Reports the year of publication when the citation is like: Corbin et al found…(year?)

Response 14: Thanks for your kind suggestion. We have changed the citation style according to your comments. (See line 59, 65 and 317)

Reviewer 2 Report

In the present study the authors have identified a cytochrome P450 gene, MoMCP1, which is required for pathogenicity and excessive Mn2+ detoxification.  Further, RNAseq results showed that Mn2+ toxin decreased the genes involved in the oxidative phosphorylation, disrupted the functions of mitochondria and vacuole, increased the ROS levels to compromise the pathogenicity and development in △Momcp1.  The paper is extensive and improves our knowledge of Mn-linked pathogeneicity.  I have only a couple of minor comments:

1.More information on the cytochrome P450 gene should be provided.  The complete gene and amino acid sequence should be presented.  Also information regarding which P450 gene family this sequence belongs to should be presented.

2.  The authors should express MoMCP1 in the △Momcp1 mutant.  If expression of MoMCP1 restores the wild type phenotype in the mutant then the authors will have conclusively proven that MoMCP1 is required for pathogenicity. At present, this essential experiment is missing.

Author Response

Response to Reviewer 2 Comments

Thanks for your kind and rapid comments, we all think it is valuable and necessary for us to improve our manuscript according to your suggestions. And the responses are following:

Point 1: More information on the cytochrome P450 gene should be provided.  The complete gene and amino acid sequence should be presented.  Also information regarding which P450 gene family this sequence belongs to should be presented.

Response 1: Thanks for your kind suggestion, it is our fault to forget to show this important sequence. And we have presented the complete gene and amino acid sequence in figure S1. We used the Fungal cytochrome P450 database ( Park, J., Lee, S., Choi, J., Ahn, K., Park, B., Park, J., Lee, Y. H. (2008). Fungal cytochrome P450 database. BMC genomics, 9(1), 402. ) to predict the MoMCP1 belonging to CYP5037B3. (See line 85)

Point 2: The authors should express MoMCP1 in the Momcp1 mutant.  If expression of MoMCP1 restores the wild type phenotype in the mutant then the authors will have conclusively proven that MoMCP1 is required for pathogenicity. At present, this essential experiment is missing.

Response 2: Thanks for your kind suggestion. We have tried several times to complete MoMCP1 in Momcp1 mutant, but all failed. Maybe, the native promoter prediction is inaccurate with no observation of fused GFP, so we decide to use the strong promoter RP27. We have transferred the RP27:MoMCP1:GFP not only in WT, but also in Momcp1mutant  successfully. Interestingly, both of overexpressed strains can’t develop appressorium, which also indicate MoMCP1 related pathogenicity. Combining with RNA-seq results, many hydrophobin proteins were regulated significantly. These results indicated MoMCP1 was related to pathogenicity. But in order to deepen function , it is necessary to  complement MoMCP1 in the Momcp1 mutant with suitable promoter in the future.

Round 2

Reviewer 2 Report

The authors have answered my points satisfactorily.